# Creating a Pharmacotherapy Collaborative Practice Network to Manage Medications for Children and Youth: A Population Health Perspective

**DOI:** 10.3390/children6040058

**Published:** 2019-04-09

**Authors:** Richard H. Parrish, Danielle Casher, Johannes van den Anker, Sandra Benavides

**Affiliations:** 1Department of Pharmacy Services, St. Christopher’s Hospital for Children–American Academic Health System, 160 East Erie Avenue, Philadelphia, PA 19134, USA; 2School of Pharmacy, Virginia Commonwealth University, Richmond, VA 23298, USA; 3Department of Pediatrics, Drexel University College of Medicine, Philadelphia, PA 19134, USA; danielle.casher@americanacademic.com; 4Universitäts-Kinderspital beider Basel (UKBB), Spitalstrasse 33, CH-4031 Basel, Switzerland; JohannesN.VandenAnker@ukbb.ch; 5Children’s National Health System, 111 Michigan Avenue, Washington, DC 20010, USA; 6Erasmus Medical Center—Sophia Children’s Hospital, s-Gravendijkwal 230, 3015 CE Rotterdam, The Netherlands; 7PRIME Education, Fort Lauderdale, FL 33309, USA; sandra.b.caballero@gmail.com

**Keywords:** clinical pharmacist, pediatrician, clinical pharmacology, pharmacotherapy, comprehensive medication management, pediatric medicines, special needs, collaboration, children, care networks

## Abstract

Children with special health care needs (CSHCN) use relatively high quantities of healthcare resources and have overall higher morbidity than the general pediatric population. Embedding clinical pharmacists into the Patient-Centered Medical Home (PCMH) to provide comprehensive medication management (CMM) through collaborative practice agreements (CPAs) for children, especially for CSHCN, can improve outcomes, enhance the experience of care for families, and reduce the cost of care. Potential network infrastructures for collaborative practice focused on CSHCN populations, common language and terminology for CMM, and clinical pharmacist workforce estimates are provided. Applying the results from the CMM in Primary Care grant, this paper outlines the following: (1) setting up collaborative practices for CMM between clinical pharmacists and pediatricians (primary care pediatricians and sub-specialties, such as pediatric clinical pharmacology); (2) proposing various models, organizational structures, design requirements, and shared electronic health record (EHR) needs; and (3) outlining consistent documentation of CMM by clinical pharmacists in CSHCN populations.

## 1. Introduction and Statement of Need

U.S. children’s hospitals and health systems have identified a lack of a standard definition for children with special health care needs (CSHCN). A lack of consensus may diminish the effectiveness of improvement projects designed to address social determinants of health in children, especially CSHCN and/or those with medical complexity (CMC) [1]. According to the Maternal and Child Health Bureau of the Health Resources and Services Administration (HRSA), CSHCN is defined as children and youth that “have or who are at risk for chronic physical, developmental, behavioral, or emotional conditions, and who also require health and related services of a type or amount beyond that by children generally.” Nearly 20% (or 17 million) of children under the age of 18 have at least one special need [2]. CMC, a sub-population of approximately 3 million, require the highest level of services and support due to their care intensity and the breadth of pediatric sub-specialists required [3]. This CMC sub-population is growing in terms of resource utilization, and requires policy and programmatic interventions that differ from broader CSHCN groups. Systems of care for CMC should be informed through evidence-based solutions to the ongoing challenges of caring for CMC with better sub-population definitions, especially in the areas of community care and outcomes measurement [4,5,6,7]. Qualitative themes have emerged in defining the population health of CMC, including health optimization and outcome measurement, and the importance of family and social determinants [8]. For community primary care providers (PCPs), patient and drug regimen characteristics such as polypharmacy, multi-organ system involvement, and rare/unfamiliar diagnoses often negatively affect care. In addition, caregivers with high needs, time constraints during visits, and challenges with care coordination were also frequently cited as impairing good clinical outcomes [9]. Key challenges are reported to be development of a consensus method to identify CMC, promotion of high-quality research, navigation of health care reform, and measures to promote professional identity and workforce development [7]. 

Within the evolving programs and systems of care for CSHCN and CMC, there is a need for better medication management processes and systems to aid in the coordination and quality of care received, especially during transitions of care and in the home environment [7,9,10,11,12]. While pediatric patients and their drug therapy needs, in general, are very different from adults, there are structures and processes of care that can be universally applied regardless of patient population. Among these are comprehensive medication management (CMM) within the structure of collaborative practice agreements (CPAs) between clinical pharmacists and physicians, such as those which have been successfully developed and implemented in adult primary care practices, associations, and health systems [13].

The overall purpose of this concept paper is to address the need for and creation of a pharmacotherapy collaborative practice network through CPAs for CMM in children and youth populations. In addition, this paper outlines the following: (1) organizing a collaborative practice for CMM between a clinical pharmacist and a pediatrician (primary care pediatricians and sub-specialties, such as pediatric clinical pharmacology); (2) describing and proposing various models, organizational structures, design requirements, and shared electronic health record (EHR) needs; and (3) outlining consistent documentation of CMM by clinical pharmacists in CSHCN populations.

## 2. Children with Special Health Care Needs and Medical Complexity (CSHCN and CMC)

### 2.1. Current Models of Care Delivery

There are three current models of care for CMC: (1) primary care-centered models; (2) consultative- or co-management-centered models; and (3) episode-based models. While these models have been evaluated favorably, they lack generalizability since standardized outcomes and population definitions for CMC are lacking. Collaboration among patients, families, providers, payers, and policy makers is required to identify gaps in care and build high reliability guidelines [6]. CMC models need to encompass comorbid neurodevelopmental conditions that frequently accompany outpatient encounters [14]. For example, CMC with upper and lower respiratory tract and gastrointestinal conditions are being treated through multidisciplinary aerodigestive centers at tertiary care centers, and integration of a general pediatrician can promote a whole-person Patient Centered Medical Home (PCMH) [15]. One report described an overarching theme “unsung heroes, flying blind” that illustrate aspects of the parents’ experience in caring for CSHCN and CMC. The enormous burden of care can decrease the parents’ ability to provide care, and impact the parents’ health, family functioning and the sick child’s potential health outcomes [16]. Comprehensive care oversight may improve care coordination but not health status for parents of CMC [17].

### 2.2. Population Health Aspects of the Patient-Centered Medical Home (PCMH)

In Figure 1, Kindig has provided a conceptual framework for understanding population health (PH), which can be applied to CSHCN and focus on both determinants of care and outcomes. Population health is defined as the “the health outcomes of a group of individuals, including the distribution of such outcomes within the group” [18]. 

Population health management (PHM) approaches to improve cost and quality have included an operating model which incorporates: “(1) a multi-layered, interprofessional infrastructure; (2) data dashboards to guide continuous quality improvement; (3) revamped provider payments; (4) consumer spending accounts; and (5) incentive alignment for shared savings with practices.” One mechanism for improving the organization and delivery of health care to populations, in general, and for CSHCN-CMC, in particular, is the Patient Centered Medical Home (PCMH) [19]. PCMH is not a place, but a model for primary care delivery encompassing comprehensive care, patient centeredness, coordinated care, accessible services, and quality and safety [20]. Primary care practices face many barriers to providing PCMH care to patients with complex needs. Reported barriers include the organization’s structure, lack of resources such as staff, space, and technology, and communication between health care team members [21]. Innovative programs that focus on patients with high resource utilization, place case managers in primary care practices, allow flexibility in matching staff to the needs of each practice, and help primary care clinicians manage CMC alongside regular patients improve care for CMC. More support is needed for smaller primary care practices to become effective PCMHs for CMC. Increased reimbursement is necessary for practices to have the resources to serve CMC effectively and to support programs that collaborate with primary care practices. Further research is required to delineate delivery models that will succeed in different types of communities [22]. Data from the 2016 National Survey of Children’s Health (N = 50 212) indicated that 43.2% of CSHCN and 50% of otherwise healthy children have access to a medical home. Attainment of the Medical Home Composite Measure, as measured by the National Survey of Children’s Health, is defined by five subcomponents including a usual source of care, having a personal physician or nurse, receiving needed referrals, receiving coordinated care, and receiving family-centered care varied significantly by sociodemographic characteristics among both CSHCN and non-CSHCN, as did attainment rates for each of the five subcomponents [23].

CSHCN with broad functional impairment (physical, cognitive, and mental health) demonstrated the highest emergency department (ED) utilization, highest odds of hospitalization, and lowest odds of a having a medical home [24]. CSHCN with functional limitations may have poorer health status and more complex health care needs, and more often experience a variety of health issues due to inadequate insurance, increased impact of health conditions on the family, and higher medical costs [25]. A third of families of CSHCN, predominantly with children from racial/ethnic minority backgrounds, those in poverty, and those with complex emotional/ behavioral or developmental needs and functional limitations, are reported to encounter difficulties, delays, or frustrations in obtaining health and related services [26,27]. Unmet needs in CMC may be double that of those without medical complexity [28]. PCMHs exist often in primary care practices, since these pediatricians are more comfortable than specialist pediatricians in providing a medical home for a child with chronic medical or developmental conditions [29,30].

In one study, CMC had a mean of 19 annual outpatient visits (USD $616) and one in four had inpatient visits ($3308), with other significant cost drivers including home health ($2957) and prescription medications ($2182). The main reasons for CMC visits were for mental health [31]. Compared to families without CSHCN, those with CMC have, on average, lower satisfaction with health care. Total spending increases for children with chronic conditions might be due to an increase in the number of recipients with the most complex chronic conditions, not increased per member per month payment to primary care providers [31]. The presence of multiple extracardiac co-morbidities is often associated with higher resource utilization following the index cardiac surgical procedure [32]. As qualifiers for special needs status among American Indian/Alaska Native (AIAN) children, the use of or need for prescription medication was the most frequent, and significantly greater disease burden among AIAN CSHCN suggests that care must be taken to ensure an appropriate level of coordinated care in a medical home to ameliorate the severity and complexity of their conditions [11].

### 2.3. Youth with Special Health Care Needs (YSHCN)

Youth with Special Heath Care Needs (YSHCN), aged 12–17 years, represent a unique sub-population of CSHCN, and a majority may have three or more comorbid conditions. Efforts for this sub-population should focus on strengthening coordinated systems of care that optimally meet the needs of YSHCN so they may thrive in their families and communities [10]. Only one-third of YSHCN had adequate transition to adult care preparation (care coordination, shared decision making, and family-centered care), and the majority that lack it are male adolescents with medical complexity [33]. One recent example of a hospital-based transition preparation infrastructure included five core objectives: “(1) facilitating medical transition with a focus on structure and processes; (2) assessing current transition policy/practice at baseline; (3) communicating the complexity and necessity of transition planning; (4) mobilizing adult providers to champion transition planning for YSHCN; and (5) identifying methods to analyze transition program building activities” [34]. PCMH can encompass such strategies in developing programs to transition YSHCN to adult care.

### 2.4. Behavioral Health

Neurodevelopmental conditions affect ~15% of US children, and the prevalence and complexity of these conditions are increasing. The workforce struggles to meet current service demands, with long waiting times for appointments, increased complexity, and high volumes of non-reimbursed care. Sustaining the subspecialty will require strategies to maintain and expand the workforce, improve clinical efficiency, and prevent burnout [35]. Older children and those living in poorer households were more likely to have an unmet need for behavioral healthcare services among CSHCN with CMC. The prevalence of unmet mental healthcare needs among both groups increased during 2005–2010 [36]. An important component of behavioral health PCMH is shared decision making (SDM), which engages the patient or caregiver in medical decisions based on the values, preferences and treatment goals of the patient or caregiver. For younger children, parents are usually involved in SDM. However, with older children and adolescents, attention deficit/hyperactivity disorder (ADHD) and/or autism spectrum disorder (ASD), most of the SDM processes involve the parent [37]. While future strategies should be implemented to incorporate the child or adolescent as appropriate, challenges due to cognitive abilities in children with CSHCN may hamper effective SDM [38].

### 2.5. Quality/Continuity of Care/Care Coordination

Care coordination is defined by the Agency for Healthcare Research and Quality (AHRQ) as “optimally organizing patient care and information-sharing activities” [39]. Coordination is an important aspect of high quality, patient-centered care. It improves safety, effectiveness, and efficiency of health care provided. Measures of coordination processes must be robust for generating evidence about its outcomes, analyzing existing practices, developing quality improvement activities, and supporting targeted payment initiatives. Continuity of care holds promise as a quality measure for CMC because of its association with lower ED utilization and more frequent receipt of care coordination [40]. Recently, the Family Experience with Coordination of Care assessed the quality of coordination as reported by the caregiver [41]. The Post-Acute Acuity Rating for Children, intended to reflect medical severity based on age, reason for admission, diagnoses, dependence in activities of daily living, and technology reliance for children admitted to post-acute care rehabilitation hospitals, has been found to be a comparable measurement of medical complexity for pediatric outpatient care [42]. Use of the CSHCN Screener, a five-item scale that identifies needs, helps to target CSHCN who need coordination, access to specialized and community-based services, and enhanced SDM. However, lack of integration into EHR limits the use of the CSHCN Screener in many clinical settings [43,44]. A greater percentage of CMC may use social work services than non-CMC, with over eight times more hours of services provided per child per year [45]. Overall family-centered care planning and delivery for CMC is optimized when the medical and social needs of the child are integrated [46]. High performing community-based complex care clinics integrated with a tertiary care center may lead to a decrease in total health care system costs per patient per month, fewer inpatient days in the tertiary care center, decreased out of pocket expenses, and durable improvements in social and emotional quality of life for children [47]. Strategies to improve coordination have included scheduling additional time with patients and caregivers, summative care plans, and facilitated synchronous and asynchronous communication with specialists [9].

## 3. Comprehensive Medication Management (CMM)

The American College of Clinical Pharmacy (ACCP) recently developed a common language document for delivering CMM [48]. Vital to communicating both internally and external to the profession, and especially to patients, the document contains the operational definitions that assure a standardized patient care process within each essential pharmacist function, as outlined below:Collect and analyze information about the patient and drug therapy history;Assess the information and formulate a medication therapy problem (MTPs) list;Develop the care plan in collaboration with healthcare team and patient/caregiver;Implement the care plan; andFollow up and monitor to optimize the care plan and resolve MTPs.

Applying implementation science principles into practice in a stepwise fashion, especially creating “bite-sized objectives” through SMART goals (Specific, Measurable, Actionable, Realistic, and Time-bound) and enlisting physician champions, has been shown to produce positive outcomes for integrating clinical pharmacists in adult primary care practices [49]. Clinical pharmacists practicing under collaborative practice agreements (CPAs) or through other privileging processes are well positioned to improve care coordination through provision of CMM. As shown in Figure 2, CMM is founded on three core components: (1) a shared philosophy of practice; (2) the patient care process; and (3) practice management systems.

In 2011, 90% of all Medicaid-insured CMC required medications that averaged US $1677 per child [50]. CMC represents about 3% of CSHCN, and CMC parents report significantly more unmet needs for prescription medications and care coordination compared to non-CMC parents. These may be associated with unmet care coordination and greater medical complexity. Pharmacists are rarely part of the CSHCN care coordination model. With the complexity of and numerous transitions of care for CSHCN-CMC, pharmacists can be responsible to advocate for the unfulfilled needs for prescription medications as care delivery models for these children evolve [12].

The American College of Clinical Pharmacy (ACCP) believes that clinical pharmacists engaged in direct patient care activities such as CMM should be board certified (i.e., residency-trained or otherwise board eligible) and have established valid CPAs or have been formally granted clinical privileges [51,52,53,54,55].

CMM is a primary method for medication optimization, a result of merging implementation science expertise with lessons learned from the “CMM in Primary Care” grant [48,53]. Six essential domains common to other health disciplines comprise CMM: “direct patient care, pharmacotherapy knowledge, systems-based care and population health, communication, professionalism, and continuing professional development.” These competencies are specifically designed to identify the unique clinical pharmacy expertise needed to provide CMM from a patient-centered, team-based perspective” [54,55]. Face-to-face CMM has resulted in increased rates of medication adherence [56]. Since 1995, the Department of Veterans Affairs has allowed Clinical Pharmacy Specialists (CPS) an expanded scope of practice with independent prescribing privileges [57]. 

Physicians have identified enhanced clinical outcomes, access to drug knowledge, and creation of a multidisciplinary model for learners as the top benefits of working with clinical pharmacists, but note limited reimbursement and billing difficulties as primary challenges [58]. Access to primary care health records can help community pharmacists build an efficient and effective CMM practice [59]. 

## 4. Collaborative Practice and Agreements (CPA)

In 2012, the American Pharmacists Association Foundation convened a roundtable consortium that generated seven recommendations for advancing pharmacists’ patient care services and CPAs which included: “use of consistent terminology;provider control over collaborative practice details;infrastructure that embeds pharmacists’ patient care services and CPAs into care;use of electronic health records and technology in patient care services;relationships among the health care team that are strong, trusting, and mutually beneficial;incentive alignments based on meaningful process and outcome measures; andredesign of health professionals’ practice acts, education curriculums, and operational policies” (see Figure 3) [60,61].

There are two primary categories of current pharmacist prescriptive authority: (1) collaborative prescribing; and (2) autonomous prescribing. Collaborative prescribing models provide a conceptual framework for treating diagnosis-established acute and/or chronic disease while autonomous prescribing models focus on a limited range of medications for minor ailments [62,63]. Practice law in four states have created advanced practice pharmacist designations, including: advanced practice (California), clinical pharmacist practitioner (Montana and North Carolina), and pharmacist clinician (New Mexico). These designations may not translate to actual scope of practice gains by comparison to other existing state laws [64,65].

To design effective community pharmacy-based point-of-care (POC) testing, development of trust between the physician and pharmacist is often based on reputation [66,67]. The most important pharmacist contributions in drug therapy that physicians have identified include adverse effect and drug interaction management, and medication access assistance education, and adherence [68]. Concise progress notes have been highlighted [69]. Examples of effective specific strategies for CPAs may include using existing physician relationships, identifying pharmacy- and regional-level champions, and allocating staffing based on prescription volume and clinical services [70]. As described for Texas federally qualified health centers, a PCMH co-visit model may induce more collaboration with physicians and more patient convenience; however, payment for the value of PCMH is not universal [71]. Physicians’ perceived barriers for collaboration have been reported to be concern over loss of communication, hesitancy to relinquish control, and lack of confidence in pharmacists’ clinical judgement [72]. Assurance of pharmacist preparedness and continuous professional development through profession-wide standards for prescribing processes will be imperative [73].

Clinical pharmacists and physicians have entered into CPAs in adult patients for at least 25 years using disease—or body system—based agreements as facilitated through state board of pharmacy and medicine collaborative practice statutes and regulations. These CPAs vary greatly from state to state; however, most regulations specify medication prescribing (initiating, modifying, and/or discontinuing pharmacotherapy), required laboratory monitoring (either venipuncture or POC), and other aspects of therapy (e.g., insurance authorizations). A CPA allows a qualified pharmacist to assess patients, order relevant laboratory tests, administer medications, and select, initiate, monitor, and adjust pharmacotherapy in collaboration with physicians and other prescribers. According to the Centers for Disease Control and Prevention, there is ample evidence that CMM enabled through a CPA effectively achieves intended outcomes [74].

CMM focuses on optimizing the patient’s entire drug therapy whereas disease management is centered on one disease state, such as diabetes, heart failure, asthma, and hypertension. In the case of CPAs involving children, CMM could be initiated in patients with two or more active chronic diagnoses that are managed with medications [75,76], for immunizations [77], during transitions of care from hospital to home [12,78], or when children with neurodevelopmental issues become adolescents [79,80,81,82].

Many jurisdictions in North America allow for pharmacists and physicians to determine the extent to which a patient is involved in a CPA, including informed consent and written consent with opt-out provisions. It is important to specify in detail the expectations and responsibilities for care. If guidelines or practice parameters are available, it is suggested that these documents form the clinical basis of the CPA. Specific definitions of drug therapy management have included adjusting a drug regimen, adjusting drug strength, frequency or route of administration, administering drugs, ordering pertinent laboratory tests, monitoring vital signs, and educating and training patients related to home management of drug therapy. A toolkit for creating CPAs for CMM in other countries is described in another paper of this Special Issue.

## 5. Network Design Requirements and Electronic Health Record (EHR) Needs

### 5.1. Overview

The speed of change from volume-based to value-based payment in health care varies widely. Focusing on the family’s well-being goals, successful community-based PH models have leveraged strong payer-provider partnerships have the potential to reduce healthcare costs and maintaining or improving care quality [83,84]. Figure 4 illustrates one PH structure where pharmacotherapy may be involved for CSHCN-CMC at several nodes in the care process. When considering the constitution of a pharmacy-based network, it is important to understand how pharmacotherapy assessment, monitoring, and follow up can be situated within the core capabilities of behavioral health, complex disease management, telehealth, and care management. Moreover, how well these core components articulate with adjacent capabilities, such as primary care networks and pediatric to adult transitions, can improve care coordination for overall medication management and positive patient outcomes.

Examples that utilize such PHM models can occur in various settings. Pharmacists have provided patient care using CPAs for diabetes, hypertension, and hyperlipidemia in PCMH clinics. Legal agreements were developed for sharing data and for accessing state-wide EHR at a large chain community pharmacy in one US state for over five years. No statistically significant differences were seen in blood glucose, lipids, or blood pressure outcomes for patients seen by PCMH versus community pharmacists, indicating that, when EHR is available, CMM can be conducted without regard to location [85]. A discharge clinic approach achieved a significant decrease in 30-day readmission rate compared with national benchmark data. The medical group’s estimated cost of readmissions was US $7,156,800, and 30-day all-cause readmission rate was 12.3%. Use of a post discharge clinic resulted in an estimated savings of $689,199 and 9.63%, a total estimated net savings of $335,199 [86]. An ambulatory care pharmacy-based transitions-of-care (TOC) program for 830 managed Medicaid patients reduced total healthcare costs at 180 days after discharge by an average of $2139 or an estimated plan savings of nearly $1.8 million [87].

### 5.2. Potential Models and Accompanying Logistical Needs

To provide for the overarching structure, potential network models are evolving. There are pharmacy-based, pediatrician office-based, telehealth (call center) based and hybrid, multi-level models. Within the pharmacy-based network, there are several existing clinically-integrated community focused networks wherein pediatric patients may receive CMM. An example of this structure is the National Association of Community Pharmacy—Community Pharmacy Enhanced Services Network (CPESN); within CPESN, all member pharmacies are capable of providing the following medication management services, including reconciliation, synchronization, vaccinations, and CMM. Additional services at select pharmacies include collection of vital signs, POC testing, administration of long-acting injections, and compounding sterile and non-sterile preparations. CPESN support solutions include quality improvement, quality assurance and practice transformation guided by CPESN clinicians and data analysts.

Some pharmacy supplier-based networks with internal clinical infrastructure can support collaborative practices (Professional Compounding Centers of America (PCCA; https://www.pccarx.com/); International Academy of Compounding Pharmacists (IACP; https://www.iacprx.org/); and Cardinal Health; https://www.outcomesmtm.com/pharmacy/medication-therapy-management/).

The infrastructure of these networks often is comprised of in-house clinical and scientific expertise that assists community pharmacists in collaboration with patients and pediatricians. For example, within Cardinal Health’s OutcomesMTM^®^ company, patients are connected through the Personal Pharmacist^TM^ Network with face-to-face and telepharmacy components to leverage pharmacists with unique access to prescribers, some of which are embedded in physician practices. Alternatively, standards-setting organizations and professional societies that have the pharmacotherapy of children as their central focus might organize collaborative and electronic networks exclusively designed to improve drug therapy outcomes in pediatrics as a population. In other words, pharmacotherapy and medication management systems for children need to include more than federal governmental approvals and pediatric society efforts; the complexities in CSHCN require increased orchestrated transdisciplinary collaboration designed to foster innovation through practice-based research and innovation.

While the number of pharmacists practicing CMM within pediatrician offices is unknown, several health system-based clinics have formed collaborations internally and with large community chain pharmacy organizations and companies to facilitate collaborative care [88,89]. In these collaborations, ambulatory clinical pharmacists are included on inter-professional outpatient teams to optimize medication use. Outpatient pharmacists conducted medication reviews while ambulatory clinical pharmacists focused on CMM. Specialty groups, such as the Childhood Arthritis and Rheumatology Research Alliance in North America and others focused on outpatient chronic care management, afford another potential network arrangement for pharmacist-pediatrician collaboration [90,91]. In a pilot study, collaboration between a community pharmacy and local pediatric primary care centers targeted asthma, a major chronic disease in children [92].

CMM can be provided via telehealth applications [93]. Primary care outcomes in a telehealth-based chronic disease management program that includes clinical pharmacists for diabetes, hyperlipidemia, hypertension control and tobacco cessation can be achieved in targeted patients [94]. Multidisciplinary therapeutic class management in pediatric behavioral health completed peer-to-peer consultations via telepharmacy for medication changes, dose reductions, and elimination of polypharmacy within or across behavioral health medication classes [79].

## 6. Consistent Documentation of CMM by Pharmacists in CSHCN-CMC Populations

### 6.1. Current Gaps in Pediatric Medication Management

A joint opinion of the ACCP Pediatrics Practice and Research Network and the Pediatric Pharmacy Advocacy Group outlined four strategies to expand the quality and capacity of pediatric clinical pharmacist care, including elevating minimum entry-to-practice expectations, standardizing education, expanding the number of pediatric clinical pharmacists, and addressing research development of pediatric clinical pharmacists and clinical scientists. The ultimate goal was to improve access for all pediatric patients, including CSHCN [95]. In addition, there are many gaps in the current system for medication management that directly affect pediatric patients and their families. These gaps can be divided into four major areas of attention: (1) lack of standardized formulations designed for children; (2) lack of information exchange and standard nomenclature to describe products used in children; (3) lack of collaboration for CMM between pediatricians and clinical pharmacists at the point of care; and (4) lack of effective feedback for understanding the impact of pharmacotherapy on patient outcomes.

Mass-produced drug products and formulations suitable for children, especially those below the age of 6 years, are lacking. Useful active pharmaceutical ingredients (APIs) with finished oral liquid dosage forms available either adapted from other countries or registered in North America and beyond has been suggested [96,97]. To optimize the effectiveness and safety of “off-label” use of compounded pharmacotherapy through complete transmission of electronic prescriptions across the continuum of care, a pediatric compounded non-sterile products repository (pCNSP) has been proposed [98]. Existing federal and corporate database structures for medical language could be employed to incorporate pCNSPs to facilitate complete transmission of electronically identifiable and standardized extemporaneous formulations. The lack of inclusion also stymies a uniform approach to CMM for pediatric patients, as these products do not appear on the patients’ medication records accurately [99]. 

Because medication use in children continues to be associated with an unacceptably high rate of adverse events, morbidity, and death, a distinct medicines-use system designed explicitly for them has been proposed [100]. A rise in the prevalence in chronic diseases such as diabetes, obesity, and inflammatory bowel diseases, is causing an increased utilization of medications that can lead to more medication errors or adverse drug events. For example, obesity now affects almost 20% of U.S. children and adolescents. Morbid obesity now affects 6% of all U.S. youth [101]. CMM in this rapidly-rising sub-population should consider for potential for drug-induced weight gain where pharmacists are embedded in the inter-professional ambulatory or community healthcare delivery for pediatric patients. Pediatric polypharmacy can be addressed through CMM services, chart reviews, and other medication-related activities [102]. Pediatric pharmacy has recently been credentialed as a board-certified pharmacy specialty through the auspices of the Board of Pharmacy Specialties (https://www.bpsweb.org) [103]. Any new network that develops for the management of pharmacotherapy in children would be bolstered with the inclusion of board-certified pediatric pharmacy specialists (BCPPS) located within health system or pediatrician group practices or within internal support staffs of community, chain, and/or supplier organizations.

### 6.2. Potential Network Architectures, Configuration, and Designs

There are two essential components of any network designed to provide patient care or services: (1) human factors related to the proximity among providers of care delivery; and (2) informatics infrastructure related to health data gathering, storage, and routing/transmission. Any design must assure that the network is stable, secure, and sustainable over time. While there are many architectures, configurations, and designs that could be employed to network CMM in pediatric pharmacotherapy, it is important that these structures be designed intentionally with the end user, children and their families, in mind. In addition, there needs to be a simulation aspect to any network so that changes in system requirements, inputs, and outputs can be tested prior to activation in a production environment.

There are two common paradigms that facilitate the distribution of information: (1) hub and spoke architectures; and (2) multicasting architectures [104]. It has been shown that providers need to connect visually and on an interpersonal basis. Applied to pediatric providers involved in this network, their proximity to each other, whether face-to-face, within the same local community, or remote / distance, is an important factor in trust development and network sustainment.

Hub and spoke designs have been applied to health care systems involving providers and informatics, academic settings, and research endeavors (see Figure 5). These involve a central unit (hub or message broker) that receives transmissions from an initiator and distributes messages to listener objects. In this structure, there may be one hub object per channel or a single master object to connect with all channels, facilitate data sharing, and SDM. In this design, the transmitter (care initiator) sends data (clinical observations and documentation, including eRx) to the network hub. The hub is comprised of listener objects (authorized care manager devices) assigned to receive information from the transmitter. The listener interprets the message and sends a new message to a terminal recipient.

Multicast involves passing synchronous messages within one or more channels that signal end users when information is available in the channel (see Figure 6). Copies are created and routed automatically to network segments that contain members of the listener object group.

In either case, network architectures for both practitioners and data can be situated in tandem or separately. For example, the Surescripts Network Alliance™ began as a functionality to facilitate electronic prescribing, and has evolved a repository for medication histories and prior authorization functionality [105]. Surescripts connects payers and pharmacy benefit managers, pharmacies, inpatient and outpatient prescribing and dispensing software, and hospitals and care providers [106]. A similar platform can be derived for a separate and distinct prescription-processing and medication management system that provides access to authorized practitioners caring for the needs of children and their families. 

### 6.3. Estimating the Number of Clinical Pharmacists Needed to Care for CSHCN-CMC

Although population estimates of the number of pharmacists needed in society is based on prescription fulfillment, data from the Bureau of Labor Statistics have shown repeatedly that there are about one-third as many pharmacists as physicians; in 2016, there were 713,800 physicians and surgeons practicing in the US [107]. According to the American Academy of Pediatrics (AAP), there were about 92,000 physician-pediatricians and over 28,000 medical and surgical sub-specialists, or about 120,000 pediatricians [108]. The AAP also believes that there is a current shortage of sub-specialty-trained pediatricians, and the distribution of generalists and primary care in rural and other underserved areas is inadequate [19]. Since estimates of CSHCN and sub-groups is elusive presently, a reasonable projection of the direct care needs of CSHCN-CMC for clinical pharmacists providing CMM in the U.S. ranges from a 1:3 to a 1:5 ratio to pediatricians, including sub-specialists. In the U.S., this would translate conservatively into between 24,000 and 40,000 clinical pharmacists needed, bearing in mind that the ratio for patients with complex chronic needs may be lower than for primary care, unless primary care is the PCMH. Moreover, the number of clinical pharmacists needed is affected by the extent to which sterile and non-sterile compounding and prescription fulfillment activities are part of the practice in addition to CMM. However, it is apparent from the preceding discussion that clinical pharmacists practicing CMM for CSHCN can improve access to safe and effective pharmacotherapies. The professions need to collaborate at local and national levels as a way of practicing in order to achieve measurable outcomes. 

## 7. Summary and Recommendations

This report contains forward-thinking statements designed to stimulate mindfulness and concerted action for improving healthcare delivery and medication management in children. Medication use is a major aspect of CSHCN, and mismanagement is fraught with ambiguity, error, and morbidity. Including board-certified pediatric clinical pharmacists providing CMM, either on-site (such as in a PCMH) or remotely, in evolving population health care networks for children has been proposed in an effort to create a dialogue that improves the likelihood for individual and population health optimization. As a medication optimization strategy, CMM has a common language and terminology for operationally defining direct patient care rendered by clinical pharmacists. Learnings from CMM application in adult primary care may afford better understanding of the potential for pharmacist collaborative care across the pediatric continuum. 

The evolution of a network of clinical pharmacists providing CMM for CSHCN may be best facilitated by insertion of pediatrics-trained and board-certified faculty from schools and colleges of pharmacy into ambulatory clinics of pediatric hospitals and health systems. Alternatively, existing pharmacist provider networks, such as the Community Pharmacy Enhanced Network and the Personal Pharmacist Network, may expand to include CMM for children based on a set of shared collaborative practice objectives. These two approaches may be integrated to form a regional network that improves access both to board-certified pediatric pharmacy specialists and to quality medications. CPAs involving children’s CMM could be initiated in patients with two or more active chronic diagnoses that are managed with medications, for immunizations, or during transitions of care from hospital to home.

## Figures and Tables

**Figure 1 children-06-00058-f001:**
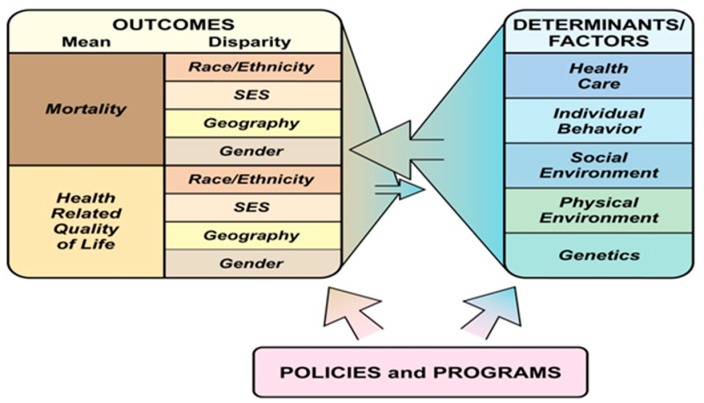
What is population health? [13].

**Figure 2 children-06-00058-f002:**
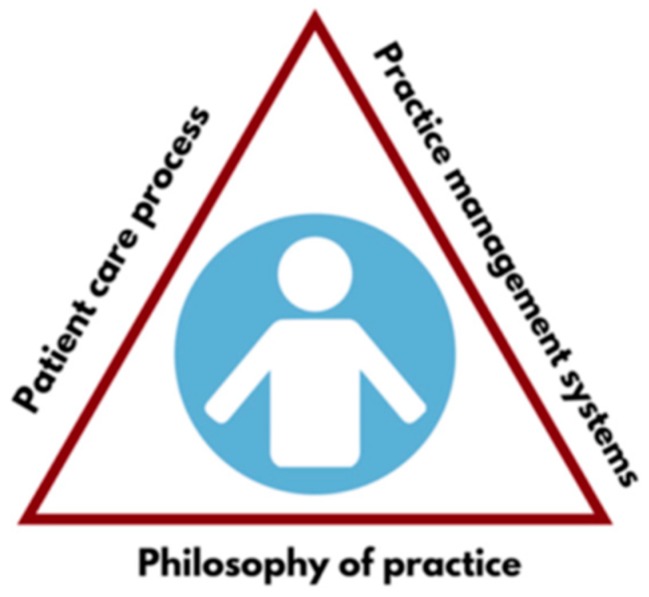
Comprehensive Medication Management framework [48].

**Figure 3 children-06-00058-f003:**
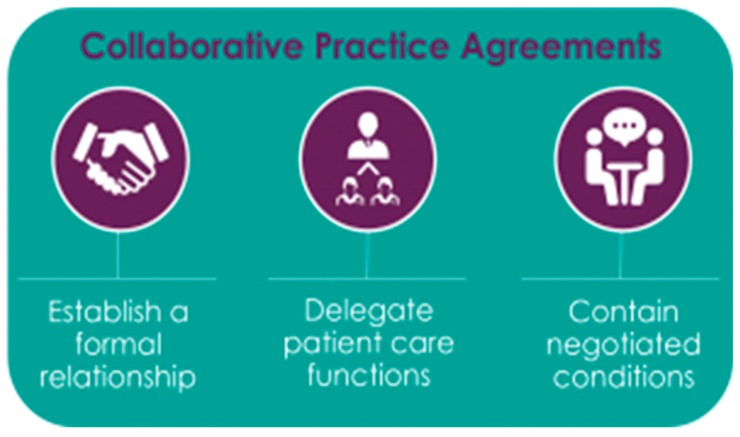
Collaborative Practice Agreement Process [60].

**Figure 4 children-06-00058-f004:**
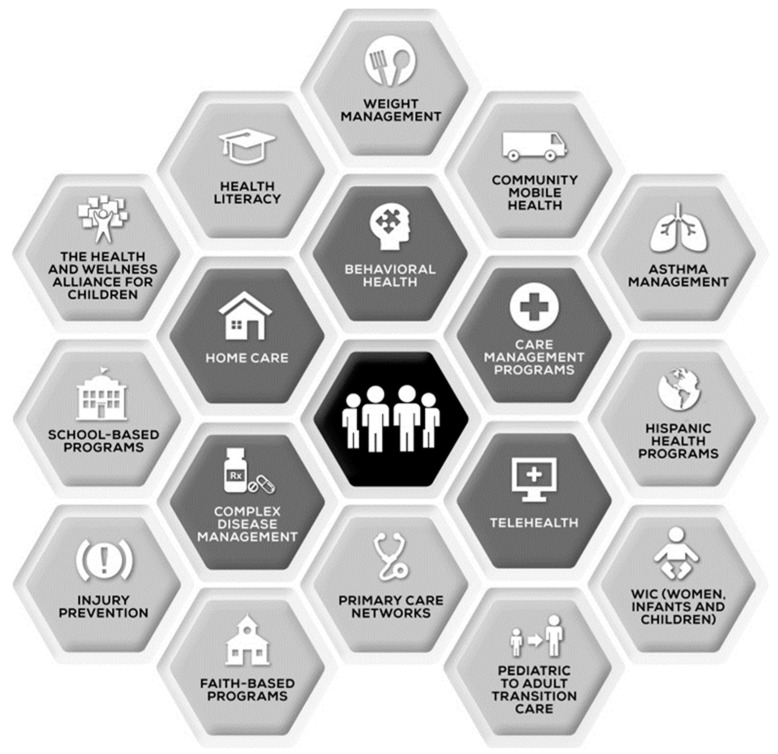
Population health (PH) capabilities [83].

**Figure 5 children-06-00058-f005:**
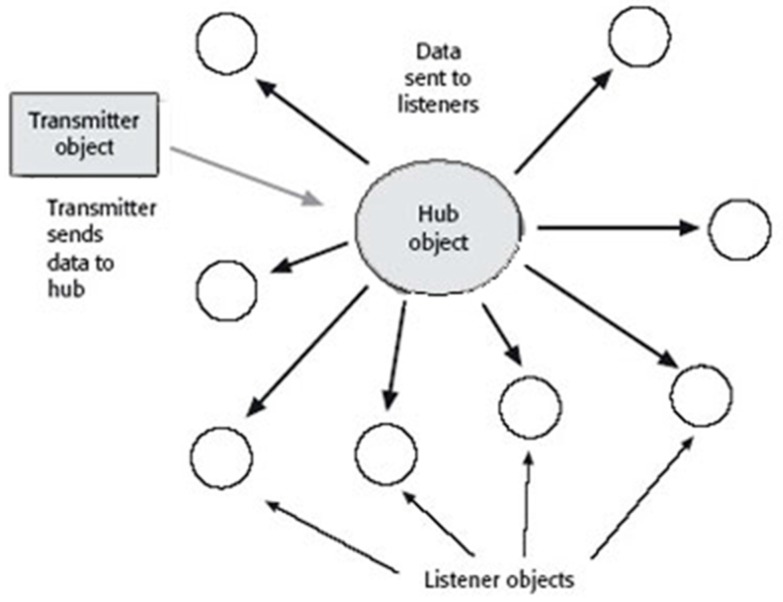
Hub and spoke architecture [104].

**Figure 6 children-06-00058-f006:**
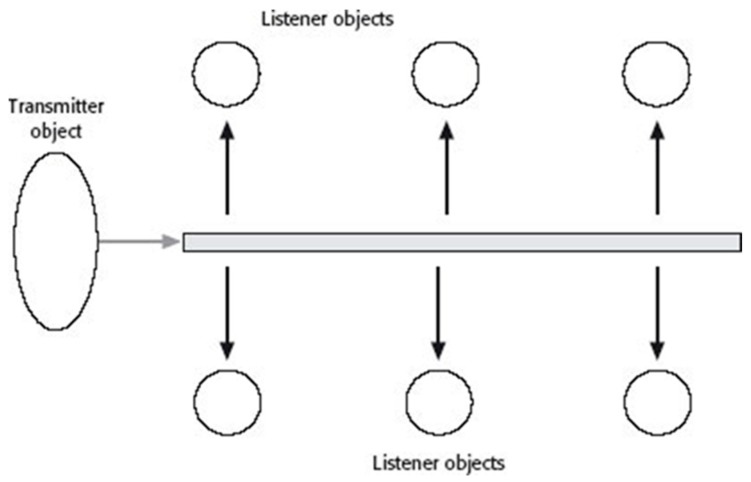
Multicast architecture [104].

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
