# Peer review of "Creating a Pharmacotherapy Collaborative Practice Network to Manage Medications for Children and Youth: A Population Health Perspective"

_children, 2019, doi:10.3390/children6040058_

Round 1

Reviewer 1 Report

This is an extremely interesting paper but its uniqueness in application to the US healthcare system limits its utility to a global audience. To make it more attractive, reference to other healthcare systems would be useful or make it less UScentric e.g costs listed in 2nd paragraph on p6 have little relevance/application outside the US, asdo the details about the roles/models in the various states (p7). Overall the content in theory is very compelling but for publication in an international journal it will require greater reference to global practice. 

Reviewer 2 Report

This manuscript deals with an important topic. It is unquestionably important that we seek ways of improving medication management for children with complex healthcare needs and it is certain that the greater involvement of pharmacists would be a worthwhile addition.

The manuscript is described as “conceptual” but it does not differentiate sufficiently among concepts nor does it provide clear advice on the approach that might work best for children. The message is obscured by the length of the manuscript and by digressions (especially between line 250 and 477) into discussion of adult care. While some of this material may be important, careful editing and focusing of message would be helpful. The inclusion of 173 references is excessive.

The paper would benefit from a greatly sharpened focus on what is promised in the title, viz, “a national pediatric pharmacotherapy collaborative practice network to manage medications”. Furthermore, the title promises a focus on children but the mss also presents some suggestions relative to youth and perhaps this could be acknowledged in a revised (and shortened) title.

In addition to the above, there are some specific issues related to grammar, syntax, language and typos, as follows.

1) Line 7

St. Christopher’s

2) line 72

“a clinical pharmacists” appears misworded

3) line 92

“The enormous burden of care can weaken the parents’ will to carry on and result in a decreased ability to provide care.” This language is flowery and non-specific.

4) line 112-114

Awkward sentence and lack of clarity “…and communication between healthcare team.”

5) line 135

missing word? mostly “those” with children?

6) line 149-154

This is a run-on sentence and should be rewritten.

7) lines 184-185

Meaning of concluding sentence unclear.

8) line 227

missing words?: totaled on average?

9) line 233

“improving” should be “meeting” unfilled needs?

10) line 247

Parenthetical comment unclear: could be omitted or “and” deleted

11) lines 276-281

A run-on sentence. Rewrite

12) line 288

Presumably these time dependent fees are hourly fees.

13) line 628

The estimates of physician-pediatrician numbers should be specifically dated

Round 2

Reviewer 2 Report

The revised manuscript is certainly more focused and generally improved although it remains lengthy and didactic in style. The ideas presented are interesting and represent a desirable new direction for (pharmaceutical) care of children and youth with complex conditions and other special needs. The information is valuable to those concerned with childcare policy.

Some of the text material could be presented advantageously in boxes or tables......for example, the list of APA recommendations presented in lines 263-270. This would help to break up the textual density. The same would be true of lists in lines 362-366 or 368-372. Particularly the latter list could be presented as a footnote or at least outside the main narrative.

While it is appreciated that the figures facilitate reading of the text, figures 2 and 3 do not add much new information. If they are to be included it is important to make sure that they are not redundant with the text.

Some minor comments:

(1) Having introduced abbreviations it is not necessary to repeat full text in lines 38 and 43.

(2)Line 53......."polypharmacy" is not a patient characteristic though patients are the victims.

(3)The phrase "most costly patient" in line 113 is pejorative.

(4)line 132......there is something missing.

(5) There is a disconnect between the observation in line 167 that 15% of US children suffer from developmental-behavioral conditions while nearly 20% (line 42) are CSHCN. Though the numbers may be correct, this seems to downplay the many other physical and genetic conditions that create special needs..

(6)Doubtful that pharmacists will take on future responsibility for "meeting unfulfilled needs for prescription medications" since one of the biggest needs is likely fiscal or policy-related in USA.

(7) line 245 "a valid CPAs"?

(8)line 260 "an efficient and CMM practice"?

Author Response

We appreciate the reviewer's comments and suggestions to strengthen the paper.

The revised manuscript is certainly more focused and generally improved although it remains lengthy and didactic in style. The ideas presented are interesting and represent a desirable new direction for (pharmaceutical) care of children and youth with complex conditions and other special needs. The information is valuable to those concerned with childcare policy.

Some of the text material could be presented advantageously in boxes or tables......for example, the list of APA recommendations presented in lines 263-270. This would help to break up the textual density. The same would be true of lists in lines 362-366 or 368-372. Particularly the latter list could be presented as a footnote or at least outside the main narrative.

We broke out the information as suggested.

While it is appreciated that the figures facilitate reading of the text, figures 2 and 3 do not add much new information. If they are to be included it is important to make sure that they are not redundant with the text.

We have checked and the information in the text expands on the figure.

Some minor comments:

(1) Having introduced abbreviations it is not necessary to repeat full text in lines 38 and 43.

We changed this text.

(2)Line 53......."polypharmacy" is not a patient characteristic though patients are the victims.

We changed the sentence to include "and drug regimen" at line 52.

(3)The phrase "most costly patient" in line 113 is pejorative.

We changed the sentence from "the most costly patient" to "patients with high resource utilization"

(4)line 132......there is something missing.

The sentence was changed from "A third of families of CSHCN are reported to encounter difficulties, delays, or frustrations in obtaining health and related services, average with children from racial/ethnic minority backgrounds, those in poverty, and those with complex emotional/ behavioral or developmental needs and functional limitations " to "A third of families of CSHCN, predominantly with children from racial/ethnic minority backgrounds, those in poverty, and those with complex emotional/ behavioral or developmental needs and functional limitations, are reported to encounter difficulties, delays, or frustrations in obtaining health and related services."

(5) There is a disconnect between the observation in line 167 that 15% of US children suffer from developmental-behavioral conditions while nearly 20% (line 42) are CSHCN. Though the numbers may be correct, this seems to downplay the many other physical and genetic conditions that create special needs..

The sentence at line 43 was changed to include "at least one special need. While rare and genetic diseases make up a significant proportion of CMC, neurodevelopmental illnesses make up a substantial group where clinical pharmacists can assist. The paragraph about costs in that section did not fit. That paragraph beginning at line 169 and ending at 181 was moved to the population health sub-section.

(6)Doubtful that pharmacists will take on future responsibility for "meeting unfulfilled needs for prescription medications" since one of the biggest needs is likely fiscal or policy-related in USA.

Great insight. The sentence was modified to change "meeting the unfulfilled needs" to "advocating for the unfulfilled needs"

(7) line 245 "a valid CPAs"?

"a" was removed from the sentence.

(8)line 260 "an efficient and CMM practice"?

"effective" was added to the sentence.